# Predicting Concentrations of Mixed Sugar Solutions with a Combination of Resonant Plasmon-Enhanced SEIRA and Principal Component Analysis

**DOI:** 10.3390/s22155567

**Published:** 2022-07-26

**Authors:** Diana Pfezer, Julian Karst, Mario Hentschel, Harald Giessen

**Affiliations:** 4th Physics Institute and Research Center SCoPE, University of Stuttgart, Pfaffenwaldring 57, 70569 Stuttgart, Germany; diana.pfezer@outlook.de (D.P.); j.karst@pi4.uni-stuttgart.de (J.K.); m.hentschel@pi4.uni-stuttgart.de (M.H.)

**Keywords:** glucose, fructose, galactose, lactose, maltose, glucose sensor, biosensing, surface-enhanced infrared absorption, principal component analysis, optical and noninvasive sensing, machine learning

## Abstract

The detection and quantification of glucose concentrations in human blood or in the ocular fluid gain importance due to the increasing number of diabetes patients. A reliable determination of these low concentrations is hindered by the complex aqueous environments in which various biomolecules are present. In this study, we push the detection limit as well as the discriminative power of plasmonic nanoantenna-based sensors towards the physiological limit. We utilize plasmonic surface-enhanced infrared absorption spectroscopy (SEIRA) to study aqueous solutions of mixtures of up to five different physiologically relevant saccharides, namely the monosaccharides glucose, fructose, and galactose, as well as the disaccharides maltose and lactose. Resonantly tuned plasmonic nanoantennas in a reflection flow cell geometry allow us to enhance the specific vibrational fingerprints of the mono- and disaccharides. The obtained spectra are analyzed via principal component analysis (PCA) using a machine learning algorithm. The high performance of the sensor together with the strength of PCA allows us to detect concentrations of aqueous mono- and disaccharides solutions down to the physiological levels of 1 g/L. Furthermore, we demonstrate the reliable discrimination of the saccharide concentrations, as well as compositions in mixed solutions, which contain all five mono- and disaccharides simultaneously. These results underline the excellent discriminative power of plasmonic SEIRA spectroscopy in combination with the PCA. This unique combination and the insights gained will improve the detection of biomolecules in different complex environments.

## 1. Introduction

Carbohydrates are essential constituents in the human diet [1], especially the sugar molecule glucose [2]. Reliable quantification of the glucose concentration in the human body is crucial for diabetes mellitus [3]. To treat the disease, continuous glucose monitoring is needed [4]. This concentration is also correlated with the interstitial fluid, human blood, or ocular fluid [5].

The detection of glucose in these complex environments is possible with infrared spectroscopy [6], which enables label-free identification and differentiation of biomolecules such as carbohydrates. Intramolecular vibrations have characteristic resonance frequencies, which lead to a distinct molecular fingerprint [7]. The problem is the small molecular absorption cross-section of mid-infrared vibrations [8]. Consequently, the glucose quantification in these complex environments is particularly challenging for low concentrations, severely limiting the applicability of standard mid-infrared spectroscopy.

One approach to overcome the limitations of infrared spectroscopy is the use of surface-enhanced infrared absorption (SEIRA) [9,10,11,12,13,14]. Plasmonic sensors are known for their high sensitivity due to the nanoscale localization of light, making them highly efficient nanoscale probes. Combining this feature with infrared spectroscopy combines the sensitivity of plasmonic sensing with the specificity of infrared spectroscopy. In SEIRA, the resonant coupling of a plasmonic and a molecular system leads to a strong vibrational signal enhancement [15,16]. Infrared light directly excites a localized surface plasmon resonance [17,18] of a metal nanostructure and thus strongly enhanced electromagnetic near-fields are generated [19,20]. When the molecules are located in these near-fields and the plasmonic resonance is resonantly matched with the molecular vibrations, the enhanced molecular vibration is seen as a modulation on top of the plasmon resonance at the characteristic resonance frequency of the molecule [21]. Plasmonic sensors are widely used in optical sensing applications such as gas sensing [22,23], biomedical sensing [24,25], and refractive index sensing [26,27] due to their high performance. As a result, optical sensors such as glucose sensors [28,29,30] can be scaled down to tens of nanometer sizes, which opens new perspectives for SEIRA applications. The use of plasmonic sensors on contact lenses as a sensing platform could provide real-time continuous and noninvasive glucose monitoring [31].

SEIRA is thus an ideal platform for the discrimination of different analytes in complex environments. The plasmonic structures offer the highest sensitivity, while the vibrational information imprinted on the far-field response allows for specificity. Combining these measurements with principal component analysis allows for an assumption-free evaluation of the data.

In this article, the detection limit, as well as the discriminative power of plasmonic nanoantenna-based sensors, is pushed towards the ultimate limit, both in terms of absolute concentrations as well as complex sensing environments. Resonant plasmonic SEIRA, in combination with machine learning, namely principal component analysis (PCA) [32], is used to predict the concentration and composition of various aqueous sugar solutions. Here, the five essential carbohydrates glucose, fructose, galactose, lactose, and maltose are probed. Our plasmonic sensor combines refractive index sensing as well as vibrational information, which is a promising tool for optical and noninvasive sensing. The excellent performance of our ansatz allows a deeper understanding of the different complex environments for future work.

## 2. Experimental Scheme

### 2.1. Vibrational Spectroscopy

Our experimental setup uses the three monosaccharides glucose, fructose, and galactose with the molecular formula C_6_H_12_O_6_, as well as the two disaccharides lactose and maltose with the molecular formula C_12_H_22_O_11_. Disaccharides consist of two monosaccharides linked by a glycosidic bond. The lactose molecule consists of a glucose and a galactose molecule, which are interlinked via a chemical bond. In contrast, a chemical bond between two glucose molecules leads to the maltose molecule. The structures of the five sugars are visualized in Figure 1a. With the aid of infrared spectroscopy, the five sugar molecules can be differentiated despite the extremely high similarity, especially between the glucose and the maltose molecule. To illustrate the distinct molecular fingerprint of the five sugar molecules in the mid-infrared region, a uniform sugar layer is applied to a bare silicon substrate by spin coating of an aqueous solution with a concentration of 100 g/L. For the five different sugar layers, Fourier Transform Infrared (FTIR) spectra are taken in transmittance, referenced to a bare silicon substrate, and depicted in Figure 1b. For each sugar molecule, an infrared spectrum is obtained. The five sugar molecules have strong vibrational bands in the wavenumber range between 1200 cm^−1^ and 800 cm^−1^. These vibrations can be assigned to the stretching modes of C-C and C-O groups of the carbohydrates [1]. The strength of the vibrational modes differs for the various sugars. Possible reasons are different film thicknesses of the sugar layer or the different magnitude of the change of the electric dipole moment during the vibration. Furthermore, the vibrational modes of the glucose and the maltose molecule are nearly at the same wavenumber, which could hamper the differentiation between these two molecules. However, the five sugar molecules exhibit many modes at different wavenumbers in the fingerprint region, and thus, identification of these remarkably similar molecules is possible via infrared spectroscopy. We would like to stress that these measurements serve as an illustration of our concept. The dried sugar films cannot be compared directly and straightforwardly in terms of sensitivity and enhancement factors to the SEIRA-based measurements in the following. The “volume concentration” in aqueous solution and in the dried film will be vastly different; additionally, dried films show generally slight spectral differences in the vibrational modes.

Due to the low molecular absorption cross-section of mid-infrared vibrations, SEIRA is utilized to investigate aqueous sugar solutions down to physiological levels. A gold nanoantenna array is used as a plasmonic sensor, which is resonantly matched with the molecular vibrations in the spectral region of interest $\omega_{\mathrm{vib}}\approx \;\omega_{\mathrm{res}}$. Here, the vibrational band of glucose at the wavenumber 1032 cm^−1^ is used as a marker band for the detection and quantification of glucose in aqueous solutions. The plasmon resonance is slightly blue-detuned to gain a higher vibrational signal enhancement [33,34]. The nanoantennas are fabricated on top of a CaF_2_ substrate via electron beam lithography.

### 2.2. Vibrational Spectroscopy—Experimental Setup

Our experimental setup is illustrated in Figure 2a. A reflection flow-cell in inverse geometry enables in-situ probing of various aqueous solutions and avoids the high absorption of water. PDMS (polydimethylsiloxane) masks provide the sealing of the flow cell. To perform spectral measurements, a commercial FTIR spectrometer (Bruker VERTEX 80, Bruker, Karlsruhe, Germany) coupled to an optical microscope (Bruker Hyperion 2000, Schwarzschild objective with 15-fold magnification, NA = 0.4) is used. Infrared light, which is polarized along the long antenna axis, excites the localized surface plasmon resonance (LSPR) of the gold nanoantenna array. Thus, enhanced and confined electromagnetic near-fields are induced. Afterward, the reflected light is measured with a liquid-nitrogen-cooled mercury cadmium telluride (MCT) detector. To tune the LSPR in the spectral region of interest, the nanoantenna has 3400 nm length, 100 nm width, and 100 nm thickness with a 2 nm chromium adhesion layer. The periodicity is adjusted to 4400 nm along the long antenna axis and to 3000 nm along the short antenna axis [35,36,37]. The corresponding SEM image of the nanoantenna array (200 µm × 200 µm) is depicted in Figure 2b. As a result, the resonant coupling between the plasmonic system and the molecule leads to an enhanced molecular vibration on top of the plasmon resonance, where the molecules have to be located inside the near-fields. A sketch of the SEIRA spectra with an aqueous glucose solution as a surrounding medium is shown in Figure 2c, and thus, the modulation appears near the characteristic resonance frequency of the glucose molecule. Moreover, the SEIRA spectra are referenced to a gold mirror.

As mentioned above, the measured SEIRA spectra are evaluated with PCA. The SEIRA spectra hold information about the concentration and composition of our probed aqueous solutions. A higher concentration of an aqueous solution leads to a higher refractive index of the surrounding medium [38]. Consequently, the plasmon resonance is red-shifted [39]. The resonance shift of the plasmon resonance enables the estimation of the concentration of the probed solution. Furthermore, the modulation on top of the plasmon resonance appears at the characteristic resonance frequency of the probed molecule. The composition of the aqueous sugar solution can be determined due to the vibrational information contained in the SEIRA spectra. Additionally, the modulation depth also depends on the concentration. A higher concentration of the aqueous solution results in a larger modulation depth because more molecules contribute to the SEIRA signal. Consequently, the desired goal is to extract these two most important pieces of information from our measured data set. One promising method to achieve this goal is the PCA, which was already successful in the identification and differentiation of pure and mixed aqueous glucose and fructose solutions [40] as well as the detection and discrimination of conformational changes of polypeptides [41]. The PCA subtracts the average *A* of all collected spectra with the total number *N* from each measured spectrum *i* ∈ [1, *N*]. Afterward, the PCA simplifies the description of the data set by decomposing the data. Each measured spectrum *i* can be described with the following equation:$${\mathrm{Spectrum}}_{i}=A+\overset{N}{\underset{j=1}{\sum }}\left({\mathrm{SC}}_{i,j}\cdot {\mathrm{PC}}_{j}\right)$$

The raw data set is now represented as a linear combination of an uncorrelated, orthogonal, and limited set of eigenfunctions and eigenvalues. The principal components (PCs) are the eigenfunctions and are the same for all measured spectra. The eigenvalues are termed scores (SCs), which are specific for each measured spectrum *i*. The integer *j* indicates the order of the PC and SCs. Furthermore, the first PC has the largest possible variance, the second PC has the second largest possible variance, and so on. The second PC is calculated under the condition to be orthogonal to the first PC. This is also the case for the other PCs. The PCs and the corresponding SCs have no a priori physical interpretation but are expected to express the concentration and the composition of aqueous sugar solutions. The physical interpretation of each PC, therefore, needs to be assisted by the spectral features which are observed. The stronger the correlation within the data set, the fewer PCs are required to express the full data set. 

## 3. Experimental Section

### 3.1. Glucose—Low Concentrations

Our experimental setup and the utilized evaluation method were introduced. The aim is to detect the glucose concentration of diabetic patients in the ocular fluid, which is ultra-low (as low as 16.6 mg/L) [31,42]. As a first step towards this goal, the sensitivity of our sensor is investigated by detecting pure aqueous glucose solutions down to 1 g/L. The following measurement cycle is performed, which is depicted in Figure 3a. Initially, the reflection cell is filled with water. Subsequently, pure aqueous glucose solutions are probed with a concentration of 1 g/L, 5 g/L, and 20 g/L. Afterward, more aqueous glucose solutions with concentrations of 5 g/L, 1 g/L, and pure water in order to prove the reproducibility of the measurement are investigated. The reflection flow cell is rinsed with deionized water to remove any residues before it is filled with a new aqueous solution. Here, 85 spectra are taken for each step, and thus, the data set has 595 spectra in total. The PCA is applied to analyze our raw data set. As a reminder, the PCs are the same for all spectra, whereas the SCs are specific for each spectrum. Consequently, 595 SCs are obtained for each order and 85 SCs for each step. First, one has to find the PC which holds the vibrational information of our measured spectra. The PCs of the second and fourth order are plotted against the wavenumber and are depicted in Figure 3b. By inspecting the second PC, maxima are present at the characteristic resonance frequencies of the glucose molecule at 1034 cm^−1^ and 1078 cm^−1^. The position of the vibrational modes varies from the previously measured FTIR spectrum in Figure 1a due to the coupling of the plasmon resonance with the molecular vibration. The second PC holds the vibrational information of our measured SEIRA spectra and hence the SCs of the second order too. In contrast, the PC of the fourth order is assumed to be noise because no physical interpretation can be found. Please note that PCA is typically used in a machine learning algorithm to reduce the dimension of a multi-dimensional problem. Yet, this typically does not imply that the PCs contain any “useful” information. It turns out for our multi-dimensional data set of glucose SEIRA spectra that the reduction of the dimensions and the data separation works best when utilizing the second and fourth order scores. Hereby, only the second PC seems to contain physically useful spectral information, whereas we cannot interpret all the other PCs. A 2D plot is generated by plotting the SCs of the fourth order against the second order, which is depicted in Figure 3c. Clusters can be observed for each measured aqueous solution. For all clusters, the mean value of the fourth order SCs is around zero and affirms the assumption that the fourth order PC is just noise. The cluster of water has the highest positive mean value of the second order SCs. By tuning the glucose concentration upwards to 20 g/L (purple), a lower mean value of the second order SCs is observed for higher concentrations. Conversely, when the glucose concentration is lowered, again, a larger mean value of the second order SCs is observed for lower concentrations. Moreover, the cluster of the same solutions, which are the aqueous glucose solutions with a concentration of 5 g/L (green and dark green) and 1 g/L (orange and yellow) as well as water (blue and dark blue), have approximately the same second order SC mean value and additionally are overlapping with each other. Hence, the reproducibility of this measurement is proven. In particular, this result implies that no sugar is sticking to the sensor. The overlap of the cluster is observed for the choice of the fourth order SCs as y-values in the 2D plot, but not for the first and third order SCs. One possible reason is the very low impact of the fourth order when compared to the first and third order. Therefore, one could just consider the mean value of the second order SCs to estimate the concentration of the different probed aqueous solutions. A one-dimensional plot works only for pure aqueous sugar solutions but not for mixed ones with two or more sugar types. This will be shown in the following. The higher the concentration of the solution, the lower the mean value of the second order SCs. Furthermore, a linear behavior is found between the mean value of the second order SCs and the concentration of the solution (see Figure 3d). The weight of the second order SCs results from the vibrational information contained in the SEIRA spectra, which was already discussed before. The linear relationship suggests that the modulation depth on the plasmon resonance scales linearly with the concentration of the solution. Here, the information about the red-shift of the plasmon resonance due to the increasing refractive index of the surrounding medium is not included in a certain order. One possibility is that the information about the resonance shift of the plasmon resonance as well as the vibrational information is included in the second order. Due to resonant SEIRA and the PCA, aqueous glucose solutions down to physiological levels can be detected. However, the limit of our sensor is already reached because the clusters of water and the aqueous glucose solution with a concentration of 1 g/L are strongly overlapping. The differentiation between the two solutions can only be conducted by the second order SC mean value of them. As a result, the sensitivity of the sensor is possibly not high enough to estimate the glucose concentration of diabetic patients in the ocular fluid. The detection of glucose is more difficult for concentrations down to physiological levels. One reason is the smaller modulation depth on top of the plasmon resonance at the characteristic resonance frequency of the probed glucose molecules because fewer glucose molecules contribute to the signal. Another reason is the small resonance shift of the plasmon resonance because of the very small concentration. Accordingly, a sensitivity down to 1 g/L is achieved by the use of a plasmonic sensor as a glucose sensor. Utilizing quantum cascade lasers sensitivities down to impressive 0.032 g/L have been demonstrated [43], which is beyond the current capabilities of our sensor. Compared to this ansatz or method allows for better localization of the sensing volume and does not require several independent laser sources. Moreover, depending on the application, one has to assess which sensitivities are required.

### 3.2. Mixed Solutions: Glucose and Fructose

So far, pure aqueous sugar solutions were investigated, but the aim is to develop a sensor that can detect the glucose concentration in the ocular fluid. In this biological aqueous environment, a multitude of different molecules are present, which complicates the glucose quantification. Hence, mixed aqueous sugar solutions are probed to predict the glucose concentration in more complex environments. From the previous results, a linear behavior of the clusters is observed when the total concentration of the pure aqueous solution is increased. Consequently, the cluster behavior of mixtures containing up to two sugars is explored by tuning the concentration of one sugar in the aqueous sugar solution. Here, glucose and fructose are utilized. The measurement cycle in Figure 4a is performed, in which 30 spectra are measured for each step. The first measured mixture is an equal mixed aqueous sugar solution with the mixture ratio G:F = 10:10. With regard to this mixture, the absolute concentration value is increased by 10 g/L of the following mixtures with only the sugar concentration tuned. First, the glucose content is increased, and afterward, the fructose content is raised. Accordingly, aqueous sugar solutions with the mixture ratio G:F = 20:10, G:F = 30:10, and G:F = 40:10 are measured as well as solutions with the inverted mixture ratio. Afterward, pure aqueous glucose solution and fructose solution with a 50 g/L concentration and an equal mixed solution with the mixture ratio G:F = 25:25 are measured for calibration. Selected raw reflectance spectra showing the plasmonic resonance as well as the imprinted vibrational features are shown in Appendix A in the Supporting Information. The raw data set is analyzed with the PCA, and thus, the obtained eigenvectors of the first, second, and third order are depicted in Figure 4b, as well as the corresponding eigenvalues in Figure 4c. The first PC looks like a plasmon resonance, and so it holds the information about the resonance shift of the plasmon resonance, which is determined by the concentration of the mixed aqueous solutions. Contrarily, the third PC contains the vibrational information of our measured SEIRA spectra because it exhibits three extrema at the vibrational modes of glucose and fructose. A maximum with a positive sign at the vibrational mode of glucose at 1034 cm^−1^ and a minimum with a negative sign at the vibrational mode of fructose at 1062 cm^−1^ is found. Additionally, a maximum with a negative sign is located at the vibrational mode of the glucose and the fructose molecule around 1079 cm^−1^. As a result, the third order SCs are plotted against the first order SCs. The total number of spectra is 630, and thus, 630 SCs are received for each order, with each measurement step containing 30 SCs of each order. The first order SCs should give information about the concentration of the aqueous solutions, and the third order SCs about the composition. The cluster of water has the highest first order SC values and a mean third order SC value around zero except for two clusters. A third order SC mean value around zero is also observed for the equally mixed aqueous sugar solutions. Furthermore, a shift of the water clusters is observed in the *x*-direction. This shift is not constant during the measurement. The reason could be stability problems of the experimental setup during the measurement. The higher the difference between the glucose and fructose content in the mixture, the smaller the third order SC value. The opposite behavior is observed for the reversed case. Therefore, the composition of the aqueous sugar solutions can be estimated. Furthermore, a linear behavior of the cluster is observed by tuning the concentration of one sugar. This enables better quantification of the composition of various mixtures. Based on the obtained results, one can assume that the modulation depth depends linearly on the concentration of the solution. In contrast, a prediction of the concentration of an aqueous solution is not possible for all probed solutions with the first order SCs. In general, the first order score value is getting lower when the concentration of the solution is increased. Moreover, the mixtures with the same concentration should have approximately an equal first order SC value. This is not the case for the equal mixed solution G:F = 10:10 (green), G:F = 25:25 (dark green), and the aqueous solution with the mixture ratio G:F = 20:10 (yellow). To achieve an accurate prediction of the concentration of the aqueous solutions, the second order will also be taken into account. The second PC has a higher impact than the third PC and minima with a negative sign are present at the vibrational modes of the glucose and fructose molecule. Therefore, the second PC can also hold information about the concentration as well as the composition of the aqueous sugar solutions. One possible reason is that the position of the modulation on top of the plasmon resonance depends on the probed molecule and the corresponding modulation depth on the concentration of the aqueous sugar solution. Consequently, the SCs of the first, second, and third order are plotted against each other. The view of the 3D plot is chosen such that no shift of the water cluster is present. As a result, eleven well-separated clusters are observed. One big cluster for all water measurements and an individual cluster for each solution are present. The cluster of water is most right-shifted. The higher the concentration of the solution, the more left-shifted the cluster of this solution. Using this fact, the concentration and the composition of the aqueous solutions can be predicted. Especially, the equally mixed solution G:F = 10:10 and the aqueous solution with the mixture ratio G:F = 20:10 can be identified. Moreover, a linear behavior is observed in this view when the total concentration of the solution is tuned. We would like to stress that there is definite potential for a better and more distinct separation for the individual measurements, possibly by utilizing higher dimensional plots. However, currently we want to demonstrate the feasibility of our method rather than its optimization. Our measurements show that several different sugar species can be discriminated, and the signals are not strongly influencing each other. Thus, we are convinced that the measurement for the small concentrations shown in Figure 3 is representative for the lower detection limit also in the case of multi-sugar sensing, which is on the order of 1 g/L. The clusters are clearly separated in this measurement, and thus, also mixtures with much lower concentrations down to G:F = 5:5 are investigated. This is depicted in Appendix A. In recent work, an adaptive method for quantitative estimation of glucose and fructose concentrations in aqueous solutions by preprocessing the data with baseline correction was developed by Schuler et al. [44]. The same method is applied for the three cases glucose and galactose, glucose, and lactose, as well as glucose and maltose. The results are depicted in Appendix A.

### 3.3. Mixed Solutions: Complex Mixtures of Five Sugars

The previous measurement has proven that accurate identification and differentiation of mixtures containing up to two sugars is possible. As a next step, the discriminative power of resonant SEIRA in combination with PCA will be investigated by probing mixtures of up to five different sugar molecules. Hence, more vibrational modes near the same wavenumber are present. One additional challenge is the differentiation between the glucose and the maltose molecule because of the extremely high similarity. The following measurement cycle in Figure 5a is performed. For all measurement steps, 30 spectra are taken. First, pure aqueous solutions with a concentration of 50 g/L are measured for the glucose (red), fructose (blue), galactose (purple), lactose (orange), and maltose (green) molecules. Afterward, mixtures of the five molecules are measured. In these mixtures, one sugar has a concentration of 50 g/L, whereas the other four sugars have a concentration of 25 g/L. Accordingly, five different mixtures are created. Selected raw reflectance spectra showing the plasmonic resonance as well as the imprinted vibrational features are shown in Appendix A in the Supporting Information. PCA is applied to analyze the raw data set, with 600 SEIRA spectra in total. As a result, 600 SCs are obtained for each order and 30 SCs for each step. In Figure 5b, the PCs of the first, second, and third order are depicted; the corresponding SCs are shown in Figure 5c. The first PC again illustrates the resonance shift of the plasmon resonance, which results from the different concentrations of the probed solutions. The third PC probably holds the vibrational information of the measured spectra, judging from the previous measurements. It exhibits three extrema in the considered wavenumber range, where also the most prominent vibrational modes of the five molecules are present. The first extremum is a minimum approximately at 1034 cm^−1^ with a negative sign and is very broad. Additionally, the vibrational modes of all the five sugar molecules are present there except the fructose molecule. The second extremum is a maximum at 1064 cm^−1^ with a positive sign, and the vibrational modes of the glucose and fructose molecule are around this wavenumber. The third extremum is a minimum, which has a positive sign and is located near the wavenumber 1077 cm^−1^. Here, the vibrational modes of the five sugar molecules are present. By inspecting the eigenvalues, the weight of the third order SCs does not enable the identification of the mixtures. Moreover, the aqueous solution with the same concentration does not have the same first order SC value. Consequently, a 3D plot of the first, second, and third order is generated like in the previous measurement. The view of the 3D plot is chosen such that all clusters of the water are overlapping. The clusters of water are most right-shifted. The higher the concentration of the solution, the more left-shifted the corresponding cluster appears. The clusters of the pure aqueous sugar solutions are well-separated from each other in the sequence fructose, galactose, lactose, glucose, and maltose. Therefore, the five sugar molecules can be differentiated, in particular the glucose and the maltose molecule, which have their vibrational modes near the same wavenumber. In contrast, the clusters of the five mixtures are not well-separated in this view. The mixture with the highest fructose content is at the top, then the positions of the cluster move downwards in the sequence of galactose, lactose, glucose, and maltose. The same sequence was also obtained for the pure aqueous sugar solutions. Despite the large number of vibrational modes near the same wavenumber range, resonant SEIRA in combination with PCA enables the estimation of the concentration as well as the identification of composition in the aqueous solutions. It should be remarked that the identification and differentiation of the solutions strongly depend on the chosen view. As a general remark, the measurements with mixtures are not reproducible because the clusters of water do not have the same SCs values. Moreover, the measurement cycle in Appendix A was performed to illustrate that also mixtures with the same concentration do not have the same SC values. Possible reasons can be stability problems during the measurement or the complexity of the data set with regard to the total number of taken SEIRA spectra and many various probed aqueous solutions. As a result, the PCA cannot recognize similar mixed aqueous solutions. In contrast, the measurement with the pure aqueous glucose solutions is reproducible. The clusters of aqueous solutions have approximately equal weight according to the mean value of the SCs. Additionally, the aqueous solutions are measured from low up to high concentrations and from high down to low concentrations. However, the mixture measurements lead to a better understanding of the working principle of the machine learning algorithm PCA in complex environments.

## 4. Conclusions 

In conclusion, we have demonstrated in-situ detection and quantification of five various sugars in aqueous solutions via resonant plasmonic SEIRA in combination with principal component analysis. The five essential carbohydrates glucose, fructose, galactose, lactose, and maltose are probed. The plasmonic sensor proved to be an appropriate glucose sensor with very high performance. Here, a sensitivity down to 1 g/L could be demonstrated. Further improvements may allow the detection of the glucose concentration in the ocular fluid. To reach a higher sensitivity, a brilliant broadband mid-IR laser [43,45,46,47] can be utilized, or different sensing materials such as graphene [48,49] or silicon [50,51] can be tried out. In addition, PCA enables the estimate of concentration as well as the composition of various aqueous sugar solutions. We observe that for complex mixtures, more than two PCs are needed for a detailed analysis. In future work, an adaptive method for quantitative estimation of the glucose, fructose, galactose, lactose, and maltose concentration in aqueous solutions by preprocessing the data with the PCA instead of the BC will be developed. Moreover, a better understanding of the PCA working principle will improve the detection of molecules in other complex environments.

## 5. Materials and Methods Section

### 5.1. Nanostructure Fabrication

The nanoantenna fabrication starts by cleaning the CaF_2_ substrate. The substrate, which is immersed in acetone, is in an ultrasonic bath at 50 °C. Afterward, the substrate is rinsed again with acetone and then with isopropanol (IPA). An O_2_ plasma clean is used to remove any small residual amount on the nitrogen-dried CaF_2_ substrate for 10 s with a power of 250 W. Moreover, the sample is spin-coated with poly(methyl methacrylate) (PMMA) as a positive tone resist. First, a 200 K PMMA with a 200 nm thick layer is used. The bottom side of the substrate is cleaned with acetone. The resist is subjected to a hard bake on a hot plate at 150 °C for 180 s. Second, a 950 K PMMA with a 50 nm thick layer is used, and the aforementioned steps are performed again. Third, an E-Spacer, is used and then the bottom side of the substrate is cleaned with water. Furthermore, the desired array of the nanoantennas (200 µm by 200 µm) is formed via electron beam lithography in the PMMA. Afterward, the sample is rinsed with water and then nitrogen-dried. Sample development is carried out due to immersion in a 1:3 methyl-iso-butylketone (MIBK) to IPA mixture for 90 s, next quenched with IPA for 60 s, and again nitrogen-dried. Accordingly, a 2 nm thick chromium adhesion layer followed by a 100 nm thick Au film is deposited with electron beam evaporation on the infrared transparent CaF_2_ substrate. Moreover, the nanoparticles are formed by a standard lift-off process. Lift-off was performed by immersing the sample in N-Ethylpyrrolidone (NEP) in connection with an ultrasound sonication at 85 °C. After a while, the sample is rinsed with acetone and then with IPA. Finally, the desired Au nanoantenna array is obtained, which is deposited on the CaF_2_ substrate.

### 5.2. Spectroscopy

All data were captured using an infrared microscope (Bruker Hyperion 2000, Schwarzschild-objective with 15-fold magnification, NA = 0.4) coupled to a commercial FTIR spectrometer (Bruker Vertex 80, Karlsruhe, Germany) with an optical path purged with nitrogen. The Globar light source emits light, which passes through a Michelson interferometer to the sample before reaching a mercury cadmium telluride (MCT) detector. An infrared wire grid polarizer was utilized to polarize the incident E-field parallel to the nanoantenna length. The sample is in a flow cell in inverse reflection geometry. An aperture size up to 50 µm by 50 µm is utilized. All infrared spectra were taken with an MCT detector under identical acquisition settings, especially 50 scans and 4 cm^−1^ resolution. The detector has to be cooled to liquid nitrogen temperatures of 77 K.

### 5.3. Flow Cell

A tailored reflection flow cell is used for probing aqueous solutions in-situ and measuring SEIRA spectra in reflection. The nanostructures are in an aqueous environment such as water or an aqueous sugar solution. All chemicals used were purchased from Sigma Aldrich. The sugar solutions were prepared by dissolving the desired amounts of glucose, fructose (F0127, Sigma Aldrich, Taufkirchen, Germany), galactose (G0750, Sigma Aldrich, Taufkirchen, Germany), lactose (61339, Sigma Aldrich, Taufkirchen, Germany), and maltose (M5885, Sigma Aldrich, Taufkirchen, Germany) in deionized water.

### 5.4. Principal Component Analysis

The measured SEIRA spectra need to be analyzed with a certain data analysis method, whereas different aqueous sugar solutions are measured. Here, it is crucial to extract both the absolute concentration value and the composition of the probed solutions. The identification and differentiation of various sugar sorts are more difficult for ultra-low concentrations down to physiological levels. One reason is the smaller modulation depth on top of the plasmon resonance at the characteristic resonance frequency of the probed molecule because fewer molecules contribute to the SEIRA signal. Another reason is the shorter resonance shift of the plasmon resonance due to smaller concentrations. The higher the concentration of the solution, the higher the refractive index of the surrounding medium. This leads to a red-shift of the plasmon resonance. The desired goal is to extract the most important information from our measured data set, which are the concentration and the composition of the solution. One promising method is the machine learning algorithm Principal Component Analysis (PCA), which was already used for the differentiation between various sugar sorts and the detection of conformational changes of polypeptides. The observations, which are contained in the data set, are characterized by certain inter-correlated quantitative dependent variables. PCA simplifies the description of the data set by representing the most important information as a set of new orthogonal variables termed principal components (PCs). The principal components are the eigenfunctions and are received as a linear combination of the original variables. The first principal component has the largest possible variance, the second principal component the second largest possible variance, and so on. The second principal component is calculated under the condition to be orthogonal to the first principal component. This is also the case for the other principal components. The values of these new variables for the observations are the factor scores (SCs), which are the eigenvalues. The factor scores are geometrically the projections of the observations onto the principal components. After applying PCA, each measured spectrum *i* can be described as follows
$${\mathrm{Spectrum}}_{i}=A+\overset{N}{\underset{j=1}{\sum }}{\mathrm{SC}}_{i,j}{\;\mathrm{PC}}_{j}$$
where *A* is the average of all measured spectra with the total number *N*. PCA subtracts the average from each measured spectra and decomposes the data. The data set is represented as a linear combination of an uncorrelated, orthogonal, and limited set of eigenfunctions and eigenvalues. The principal components are the same for all measured spectra, and the scores are specific for each measured spectrum *i*. The correlation of the data set is more notable when fewer principal components are required in order to describe the data set. Hence, the pattern of similarity of the observations and of the variables are displayed as points in maps. Moreover, the principal components and the corresponding scores have no a priori physical interpretation but are expected to express the concentration and the composition of aqueous sugar solutions due to PCA.

## Figures and Tables

**Figure 1 sensors-22-05567-f001:**
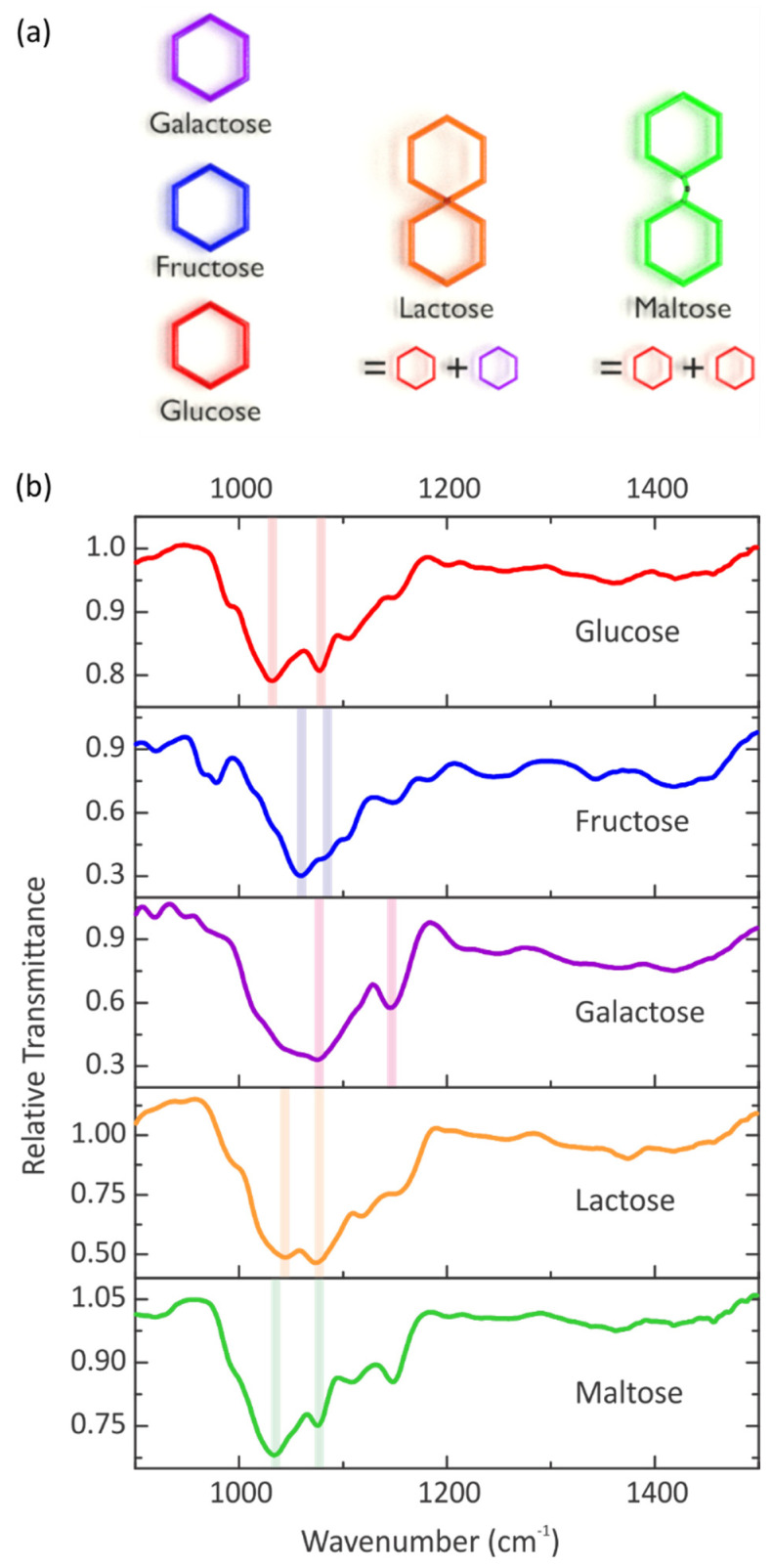
(**a**) Artistic sketch of the structures of the three monosaccharides glucose, fructose, and galactose, as well as the two disaccharides lactose and maltose. The lactose molecule is composed of a glucose and a galactose sub-unit, while maltose consists of two glucose sub-units. The disaccharides are thus expected to have similar vibrational spectra when compared to the individual sub-units, rendering their discrimination challenging. (**b**) Relative transmittance spectra of five sugar layers. The layers are spin coated from an aqueous sugar solution with a concentration of 100 g/L on a silicon substrate. The spectra are referenced to a bare silicon substrate to show the individual vibrational modes of glucose, fructose, galactose, lactose, and maltose, where the lines emphasize the most prominent vibrational modes.

**Figure 2 sensors-22-05567-f002:**
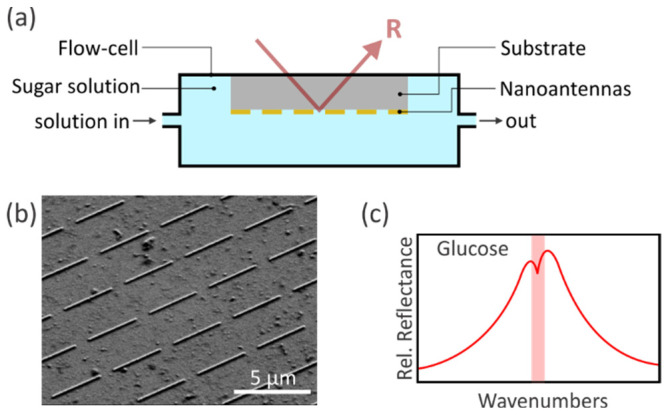
(**a**) Reflection flow-cell for FTIR spectroscopy in aqueous environments. Tubing allows to flush the desired aqueous solution through the flow cell. The nanoantennas are fabricated on top of an infrared transparent CaF_2_ substrate and are immersed in the aqueous environment. Incident infrared light excites the plasmonic resonances of the nanoantennas, which are resonantly matched with the vibrational modes of the five sugar molecules. The SEIRA spectra are obtained by the detection of the reflected light. (**b**) SEM image of the gold nanoantenna array (200 µm × 200 µm). The nanoantenna length is 3400 nm with a period of 4400 nm in *x*- and 3000 nm in *y*-direction. The *x*-direction (*y*-direction) is along the long (short) antenna axis. The width and thickness are adjusted to 100 nm. Additionally, a 2 nm chromium adhesion layer is on top of the CaF_2_ substrate. (**c**) Sketch of the enhanced molecular vibration of the glucose molecule on top of the plasmon resonance in relative reflectance.

**Figure 3 sensors-22-05567-f003:**
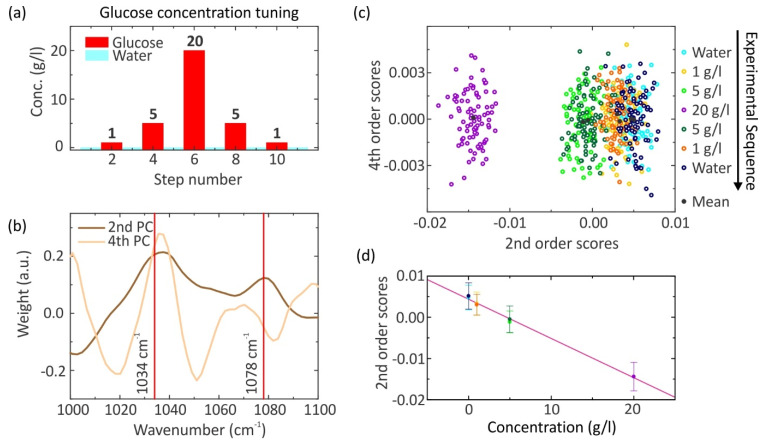
(**a**) Measurement cycle of very low concentration aqueous glucose solutions and water. The red bars indicate the glucose concentration of the solution; 85 spectra are taken for the aqueous glucose solutions as well as for water in the first and eleventh step. (**b**) The principal components of the 2nd (brown) and 4th (sandy brown) order are plotted against the wavenumber. The red lines indicate the vibrational modes of glucose at 1034 cm^−1^ and 1078 cm^−1^. As can be seen from the spectral behavior, the 2nd PC contains vibrational information. (**c**) The 4th order scores are plotted against the 2nd order scores. The measurement cycle is depicted on the right and runs from top to bottom. All measurement steps are color-coded. The mean value of the 2nd order scores enables the concentration estimation of the aqueous glucose solution down to a concentration of c = 1 g/L. (**d**) Mean 2nd order scores as a function of glucose concentration, extracted from (**c**) demonstrating the linearity of the measurement. The color code matches the color code in (**c**). A linear fit is plotted in red. The error bar represents the standard deviation.

**Figure 4 sensors-22-05567-f004:**
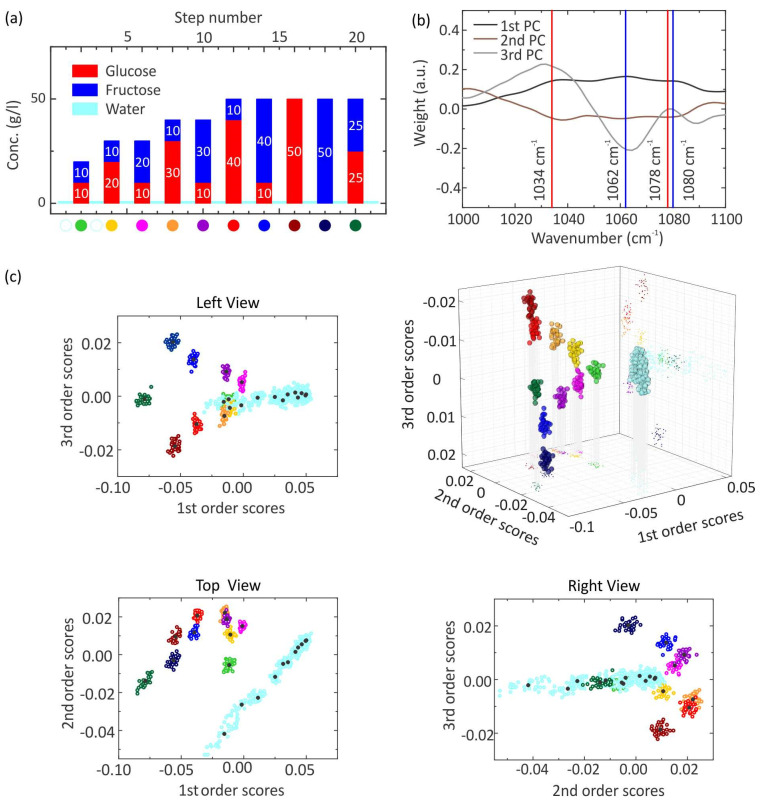
(**a**) Measurement cycle of pure and mixed aqueous sugar solutions of glucose and fructose as well as water, where 30 spectra are taken for each step. The red bars indicate the concentration of glucose in the solution, the blue bars of fructose, and the cyan bars of water. (**b**,**c**) PCA of the measured spectra from (**a**). (**b**) The PCs of the 1st, 2nd, and 3rd order are plotted against the wavenumber. The 3rd PC (gray) contains the vibrational information. The red lines indicate the vibration mode of glucose at 1034 cm^−1^ and 1078 cm^−1^ and the blue lines of fructose at 1062 cm^−1^ and 1080 cm^−1^. (**c**) A 3D plot of the 1st, 2nd, and 3rd order scores is generated as well as the corresponding 2D plots. Each measurement step is color-coded, where water has the same color and the other solutions have an individual color that is indicated at the bottom of the measurement cycle in (**a**).

**Figure 5 sensors-22-05567-f005:**
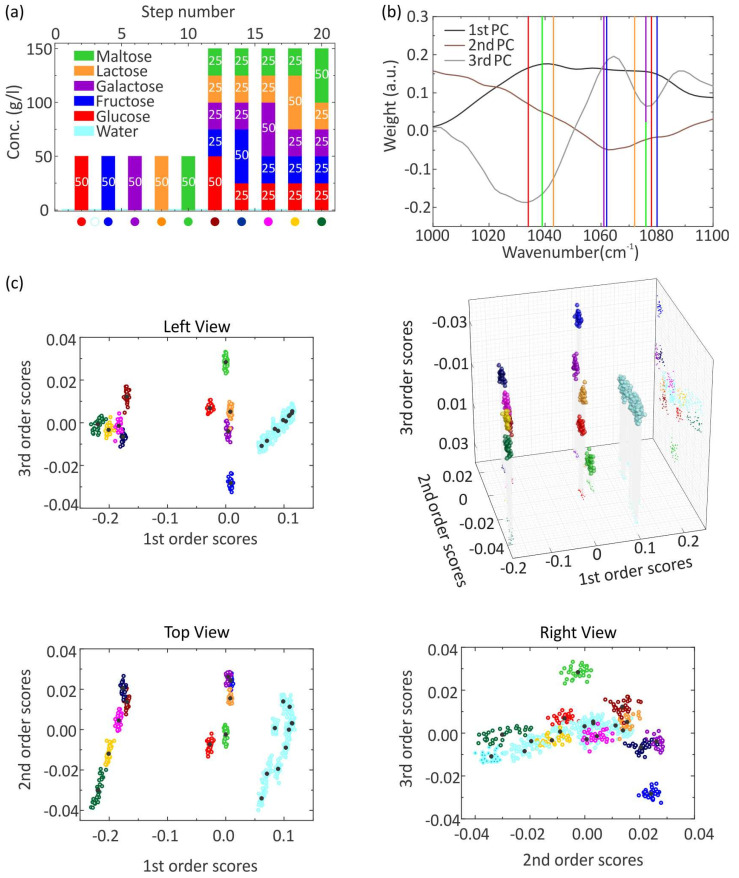
(**a**) Measurement cycle of pure and mixed aqueous sugar solutions of glucose, fructose, galactose, lactose, and maltose as well as water, whereas 30 spectra are taken for each step. The bars indicate the concentrations of glucose (red), fructose (blue), galactose (purple), lactose (orange), and maltose (green). The cyan bars indicate a pure water solution. (**b**,**c**) PCA analysis of the measured spectra of (**a**). (**b**) The PCs of the 1st, 2nd, and 3rd order are plotted against the wavenumber. The 3rd PC (gray) contains the vibrational information of the molecules. The red lines indicate the vibration mode of glucose at 1034 cm^−1^ and 1078 cm^−1^, the blue lines of fructose at 1062 cm^−1^ and 1080 cm^−1^, the purple lines of galactose at 1043 cm^−1^, 1061 cm^−1^, and 1076 cm^−1^, the green lines of maltose at 1039 cm^−1^ and 1076 cm^−1^, the orange lines of lactose at 1043 cm^−1^ and 1072 cm^−1^. (**c**) The SCs of the 1st, 2nd, and 3rd order are plotted against each other to create a 3D plot. Moreover, the corresponding 2D plots are generated. For each solution, a cluster is obtained, which has an individual color depicted at the bottom of the measurement cycle in (**a**). The measurements with water have the same color.

## Data Availability

Not applicable.

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
