# Peer review of "Predicting Concentrations of Mixed Sugar Solutions with a Combination of Resonant Plasmon-Enhanced SEIRA and Principal Component Analysis"

_sensors, 2022, doi:10.3390/s22155567_

Round 1

Reviewer 1 Report

The manuscript reports on a novel method for the detection of glucose and other sugars in solution based on SEIRA and PCA. The sensing strategy proposed by the authors is interesting and quite innovative, nevertheless there are several criticisms that need to be addressed to make the manuscript suitable for publication.

A list of the main criticisms is reported in the following:

1. The title is misleading since it states that the method employed to predict the sugar concentrations is based on machine learning, but the analysis reported in the manuscript are based on PCA that is a data classification technique. On the contrary, machine learning algorithms require a learning system, usually performed by training the algorithm with a model data set, that allows to make predictions on unknown data.

2. The manuscript lacks of a material and method section in which details on sample preparation, experimental measurements (instruments and measurements parameters) and data analysis (data processing, employed software or algorithms) are clearly reported.

3. The text refers to Supplementary Figures (for example Figure S3-S5 at line 346) but Supplementary Materials have not been provided with the manuscript.

4. Authors should report the resonance wavelength of the plasmonic antenna together with the plasmonic resonance spectrum. This aspect is relevant since authors state that the weights corresponding to the 1st PC in Figure 4 and 5 looks like the plasmonic resonance of the antenna.

5. Authors should report in the Supplementary material a comparison between the spectra of Figure 1b and the SEIRA spectra of the sugars acquired by employing the plasmonic nanoantenna. Moreover, an estimate of the SEIRA enhancement factor would be appropriate.

6. Given the linear relationship between the scores of Figure 4c, is it possible to extrapolate a limit of detection for the glucose concentration.

7. Which are the advantages of the proposed methodology in terms of limit of detection and specificity with respect to similar systems present in literature?

 Minor points

1. At line 203, authors report that the glucose concentration in the ocular fluid of diabetic patients in ultra-low. Please add a range of values according to literature.

2. The colors reported in the text for the discussion of the data of Figure 1c (lines 228, 232) do not correspond to the color code employed for the concentrations in the Figure.

Reviewer 2 Report

Authors demonstrated in-situ detection and quantification of five various sugars in aqueous solutions via resonant plasmonic SEIRA in combination with principal component analysis (PCA). The results are interesting, Authors combine experimental techniques with machine learning (PCA) which is very practical. I think it can be accepted for publication after minor revision:

The article can be accepted for publication after minor revision:

-      - I would re-write the title of the manuscript to:

“Predictions of Concentrations of Mixed Sugar Solutions with Combination of Resonant Plasmon-Enhanced SEIRA and Machine Learning”

-     -  SEIRA technique must be better described in the manuscript (why this technique was chosen?)

-     - I think Authors should elaborate more on the results in terms of other techniques: what are the advantages of using Authors’ method?

Reviewer 3 Report

The Manuscript Combination of Resonant Plasmon-Enhanced SEIRA and Machine Learning Predicts Concentrations of Mixed Sugar Solutions Down to Physiological Levels presents a method for glucose determination in water solution at low concentrations based on the combination of SEIRA and PCA. The work is sound and well written; the ability to distinguish sugars in a complex mixture with attention to their concentration is of great interest for applications (also given interesting possible improvements depicted in the conclusions). So, I recommend publication once addressing some minor points are addressed:

  1. For the experiment performed with glucose at low concentrations, it is stated that 2nd order PC contains the vibrational information while 4th order PC is essentially noise; any idea about the kind of information brought by 1st and 3rd order PCs? Which percentage of variability does each PC express?
  2. Pag. 7, line 239: Score separation correlates well with concentration, and the reproducibility of the measurements is also good. However, the scores are relatively dispersed and significantly overlapping, so I suggest talking about "estimation" rather than "determination" of concentration. Consider smaller concentration intervals for subsequent subsequent work.
  3. Pag. 7, lines 242-244: I suggest showing more clearly the linearity plotting mean 2nd order SCs vs concentration.
  4. Pag. 8, lines 285-287: It is not clear in the text what the concentrations of the solutions are at the different compositions. Have you tried to consider a fixed total concentration and the same glucose/fructose reciprocal ratios? Do you still get a good separation?
  5. Pag. 9, lines 315-316: this is strictly true only for fixed glucose and varying fructose concentrations, whereas in the series with fixed fructose, the glucose concentrations at 30-20-10 g/l are almost overlapping with each other and with water. Perhaps plotting mean value SCs vs composition might make the distinction clearer?
  6. Pag. 9, lines 325-327: except that around 1034 cm-1, the minima in 2nd PC are not as evident, at least for fructose, in the scale shown in fig 4b
  7. Figure 3: check for the c) label in the caption
  8. Figure 4: the colors of the curves in panel 4a are very similar hindering readability. Making the color code in panel 4a perfectly match 4c would improve the readability of the graph as would, in my opinion, avoid repeating the water label at each step. Perhaps you could consider a 4-entry chart with glucose concentration, fructose concentration, total solution concentration and label color instead of the chart in 4a.
  9. Figure 5c: yellow and orange labels in the 3D plot are not distinguishable  
  10. the figure s1 mentioned in the main text cannot be found, please provide

Round 2

Reviewer 1 Report

The authors addressed all the raised issues.

The manuscript is now suitable for publication.